# Development of a Disease and Pest Management Program to Reduce the Use of Pesticides in Sweet-Cherry Orchards

**Manuel González-Núñez** [1], **Pilar Sandín-España** [2], **Miguelina Mateos-Miranda** [2], **Guillermo Cobos** [1], **Antonieta De Cal** [1], **Ismael Sánchez-Ramos** [1], **Jose-Luis Alonso-Prados** [2,*] and **Inmaculada Larena** [1]

1    Department of Plant Protection, Instituto Nacional de Investigación y Tecnología Agraria y Alimentaria (INIA-CSIC), Carretera de La Coruña Km. 7, 28040 Madrid, Spain
2    Unit of Plant Protection Products, Instituto Nacional de Investigación y Tecnología Agraria y Alimentaria (INIA-CSIC), Carretera de La Coruña Km. 7, 28040 Madrid, Spain
*    Correspondence: prados@inia.csic.es

**Abstract:** A protocol for managing the main diseases and pests of sweet cherry in Spain (New IPM) has been implemented in order to reduce the use of pesticides. This New IPM includes nonchemical strategies, such as biological products against diseases and mass trapping of pests, and adjusts the timing and number of pesticide applications according to damage thresholds and a predictive model of diseases based on climatic factors. The New IPM was compared—in commercial orchards from the main cherry-producing areas in Spain (Aragon and Extremadura)—to the integrated management usually carried out in these areas (Standard IPM). Furthermore, a multiresidue method for the determination of the residues in cherries was developed. The number of applications, active ingredients used, and residue levels in fruit were reduced very significantly with this New IPM without affecting the effectiveness in the control of the main cherry diseases (brown rot, shot-hole, and leaf-spot) and pests (European cherry fly, spotted wing drosophila, and black cherry aphid). Neither significant differences in the abundance and diversity of microorganisms in flowers and fruit nor soil and canopy dwelling arthropods were observed between the two protocols, although some positive effects of this New IPM were seen on some groups of natural enemies.

**Keywords:** forecasting models; cherry diseases; cherry pests; mass trapping; pesticide residues

## 1. Introduction

Spain is the fifth-largest cherry area in the world, behind Turkey, Chile, the USA, and Italy [1]. Cherry trees occupy more than 27,500 ha in Spain, where Extremadura and Aragon account for more than 60% of the area, with a cherry production of 36,233 and 32,859 t, respectively [2]. Diseases and pests affect cherry production in both areas [3,4].

The most important cherry diseases caused by fungi in Spain are brown rot (BR), shot-hole (CSH), and leaf-spot (CLS), caused by *Monilinia laxa* (Aderh. & Ruhland) Honey, *Stigmina carpophila* (Lév.) M.B.Ellisand, and *Blumeriella jaapii* (Rehm) Arx, respectively. BR causes blossom blight and fruit rot, where losses in cherries can be up to 33% at harvest, and after cold storage at 0 °C for 1 month, losses of up to 86% due to rotted fruit have been reported [5]. CLS causes defoliation by midsummer, which results in soft, poorly colored fruit that is low in soluble solids [6], delays acclimation of fruit buds and wood to cold temperatures in the fall, increases tree mortality during severe winters, and reduces fruit-bud survival and fruit set the following year [7]. CSH causes brown lesions on leaves surrounded by a light-green margin, which eventually detach, giving a peeling or tearing effect. On fruit, CSH causes the same lesions, which eventually become rough and suberous. Significant infections can decrease fruit production. On shoots and branches, CSH lesions evolve towards the formation of cankers with a gummy exudate around them [8]. Climate factors (temperature (T), the relative humidity (RH), wetness duration (W) and the amount of rainfall (R)) affect cherry diseases [9–11]. Forecasting models can assist producers in

estimating the possibility of cherry diseases based on climatic factors and in selecting and timing preventive applications [11–13].

The most damaging pests to cherry trees in Spain are the European cherry fly (*Rhagoletis cerasi* (L.) (Diptera: Tephritidae)), the spotted wing drosophila (*Drosophila suzukii* (Matsumura) (Diptera: Drosophilidae)), and the black cherry aphid (*Myzus cerasi* (F.) (Hemiptera: Aphididae)). Females of *R. cerasi* lay their eggs under the cherry cuticle and larvae feed in the mesocarp, totally depreciating the fruit [14]. The tolerance level for the market is only 2% of fruit infested by *R. cerasi*, and in the absence of adequate control, all production may be lost [15]. The spotted wing drosophila is an invasive species originally endemic to South Asia that arrived recently to Europe and North America, where it has become a serious problem for small fruit crops. Females of this species are able to lay eggs in healthy ripening fruit of a wide variety of cultivated and wild fruit species, the cherry tree being one of the crops in which it produces the greatest damage [16]. Depending on the crop, yield losses have been estimated to reach 80% [16,17]. The control of this pest offers great difficulty, since it has high fecundity and development speed, good adaptation to adverse environmental conditions, great dispersal potential and numerous cultivated and wild hosts [17]. The black cherry aphid (*M. cerasi*) is a cosmopolitan species that colonizes plants from a few families and can cause great damage to cherry trees. It establishes large colonies in the undersides of leaves of growing shoots, curling and distorting the leaves and suppressing tree growth, and secretes a honey dew that can cover the fruit, making it unsellable [18].

Cherry diseases and pests have traditionally been treated with pesticides, with a predominantly contact or systemic effect, during the growing season from early spring until close to harvest. In consequence, monitoring the chemical pesticide residues levels in cherry fields is crucial to ensure fulfill the maximum residue levels (MRLs).

Rapid and reliable methods are essential to ensure that the residual levels in cherry fruit are under the cited safety levels (MRLs) established by the European authorities [19,20]. For monitoring pesticide residues in food commodities, multiresidue methods are highly preferable due to the simplicity of detecting several analytes in a single extraction and analysis step, facilitating the demands of more efficient monitoring of pesticides [21,22]. However, it is also desirable to minimize the use of plant protection products and their risk by 50% to fulfill the objectives of the "farm-to-fork" strategy [23].

The European Commission published in 2020 the "farm-to-fork" strategy [23] that will enable the transition to a sustainable food system that safeguards food security and ensures access to healthy diets from a healthy planet. It will reduce the environmental and climate footprint of the food system and strengthen its resilience, protect the health of citizens, and guarantee the livelihood of economic operators.

The aim of this study was to investigate the possibilities for reducing pesticide treatments on cherry ecosystems and environment. With this objective, field studies were carried out comparing both efficacy and pesticide levels in fruit of the integrated management for pests and diseases, usually applied by cherry producers (Standard IPM), with a new integrated management program (New IPM). This New IPM was implemented including nonchemical control strategies, adjusting pesticide application times, and adapting the Standard IPM to possible future restrictions on the use of some pesticides. In addition, a multiresidue method, based on QuECheRS and LC-ESI-MS/MS, was developed to compare pesticide residue levels in cherry fruit in both the Standard IPM and the New IPM.

## 2. Materials and Methods

### 2.1. Field Trial Orchards

For field trials, cherry orchards representative in the main producing areas of cherry in Spain (Aragon and Extremadura) were selected (Table 1). On the right bank of the river Ebro (Aragon), four orchards were selected and divided into two plots of at least 400 m$^2$, and one of the two management programs (New IPM and Standard IPM) was carried out in each of these plots. In the Jerte Valley, the main cherry-producing area of Extremadura, the

size of the cherry orchards is generally insufficient to apply both management programs in the same orchard. Therefore, four pairs of orchards were selected and each orchard within a pair was considered a plot (with at least 400 m$^2$ each orchard). Each IPM program was carried out in one of the plots of the pair and with the plots of each of these pairs as close as possible and similar in environmental features and cherry varieties.

**Table 1.** Location and cultivars of cherry orchards selected for comparison trials of disease and pest protocols.

| Area | Orchard | Management | Location (x, y) | Cultivar |
|---|---|---|---|---|
| Aragon | La Loma S<br>La Loma N | Standard IPM<br>New IPM | 41.5097, −1.2924 | Sonata/Satin<br>Sonata/Satin |
| | La Cuesta S<br>La Cuesta N | Standard IPM<br>New IPM | 41.4820, −1.3492 | Satin/Frisco<br>Satin/Frisco |
| | El Saso S<br>El Saso N | Standard IPM<br>New IPM | 41.4952, −1.2817 | Early Bigi/Nimba<br>Early Bigi/Nimba |
| | Barrachina S<br>Barrachina N | Standard IPM<br>New IPM | 41.4516, −1.3997 | 3–13/Prime Giant<br>3–13/Prime Giant |
| Extremadura | Collado<br>Albarizas | Standard IPM<br>New IPM | 40.1391, −5.9295<br>40.1477, −5.9230 | Van/Lapins<br>Van/Lapins |
| | Cañadilla S<br>Cañadilla N | Standard IPM<br>New IPM | 40.1208, −5.9410<br>40.1207, −5.9385 | Celeste/Lapins/Early Lory<br>Early Lory/Lapins |
| | Cerro S<br>Cerro N | Standard IPM<br>New IPM | 40.1412, −5.8869<br>40.1408, −5.8850 | Early Billy/Lapins/3–13<br>Burlat/Early Lory |
| | Porras S<br>Porras N | Standard IPM<br>New IPM | 40.1965, −5.7752<br>40.1968, −5.7755 | Lapins/Van/4–84<br>Lapins/Van/4–84 |

*2.2. Developing a New IPM for Sweet Cherry Diseases and Pests*

The tools available for prevention and control of the main diseases and pests in the Spanish cherry orchards are listed in Table 2.

**Table 2.** Available tools for managing cherry diseases and pests in Spanish cherry orchards.

| Available IPM Tools | |
|---|---|
| Monitoring | *Monilinia* spp.: Numbers of mummies, damaged buds, necrotic flowers and rotten fruit<br>*Myzus cerasi*: Hatched eggs (%) in buds swelling stage and shoots occupied (%) postharvest<br>*Rhagoletis cerasi* and *Drosophila suzikii*: Adults/trap and % damaged fruit<br>*Tetranychus urticae*: leaves occupied (%) by *T. urticae* and Phytoseiidae |
| Cultural practices | Removal of *Monilinia* mummies and branches affected by fungi<br>Proper pruning and disinfection of pruning tools<br>Balanced fertilization and irrigation<br>Remove and destroy unharvested fruit<br>Avoid water stress and dust on the leaves |
| Physical methods | Sticky barrier on trunks to prevent ants from protecting and transporting aphids on trees |

**Table 2.** *Cont.*

| Available IPM Tools | |
|---|---|
| Biological control | *Bacillus subtilis* (against *Monilinia* spp.)<br>*Bacillus thuringiensis* (against Lepidoptera)<br>*Beauveria bassiana* (against *R. cerasi*)<br>Phytoseiidae (against *T. urticae*)<br>Conservation and enhancement of specific and generalist natural enemies |
| Other nonchemical methods | Mass trapping (against *R. cerasi* and *D. suzukii*) |
| Chemical control | Abamectin, acetamiprid, acrinathrin, azadirachtin, cyantraniliprole, cypermethrin, deltamethrin, esfenvalerate, flonicamid, formetanate, hydrolyzed protein, λ-cyhalothrin, spinetoram, spinosad, spirotetramat, sulfoxaflor, tau-fluvalinate Boscalid, calcium polysulfide, captan, copper hidroxide, copper oxide, cuprocalcic sulfate, difenoconazole, dodine, fenbuconazole, fenhexamid, fenpyrazamine, fluopyram, pyraclostrobin, tebuconazole, tribasic copper sulfate, trifloxystrobin |

Mathematical models have been developed to estimate BR [11], CSH [13], and CLS [12] based on climatic factors. The climatic factors used in these models are mainly based on T, RH, W (h), R (mm) and number of days with high humidity or precipitation (days). To validate forecasting disease development, real climatic factors were measured by an automated data logger (EL-USB-2, Lascar electronic, UK) placed in each New IPM plot at a height of 1.5 m above the ground. Hourly readings of T (°C) and RH (%) were recorded by a mobile weather station installed from March to June, 2021. In addition, data on T (°C), RH (%), and R (mm) from two stationary weather station in Valdastillas (40.1412, −5.8688 at 515 m altitude and 3–10 km from the experimental plots in Jerte Valley, Extremadura), and Epila (41.5986, −1.2797 at 336 m altitude and 3–10 km from the experimental plots in Aragon) were collected. Using these data, the daily average T, RH, and R from flowering to harvest were calculated using Microsoft Excel. The months of March and April were divided into fortnights, for which the mean values of T, RH, R, and number of days in which the RH was higher than 80% were calculated. These four fortnights of March and April were considered the critical period for cherry disease.

In the New IPM proposed for the main arthropod pests, prevention is also prioritized over control measures. Among control measures, those that are not chemical are applied in the first instance. Thus, mass trapping was carried out preventively against *R. cerasi* and *D. suzukii* in those New IPM plots with a history of high incidence of these pests (Barrachina and Cañadilla against *R. cerasi* and Barrachina and Albarizas against *D. suzukii*). Insecticides were only applied when preestablished thresholds were exceeded, giving priority to the most selective and least residual products. In the case of the European cherry fly, bait applications were preferred to spraying the entire foliage.

*2.3. Disease and Pest Monitoring/Assessment*

BR, CSH, and CLS incidence (%) were monitored monthly on 10 cherry trees in each cherry plot in both areas, from flowering to harvest. Each disease's incidence was measured as the percentage of diseased leaves or fruit per tree and plot at each monitoring day. The area under the disease progress curve (AUDPC) was calculated for each IPM and plotted on an Excel spreadsheet using the formula [24]:

$$AUDPC = \sum_{i=1}^{n-1} \left[ \frac{(t_{i+1} - t_i)(v_i + v_{i+1})}{2} \right]$$

where *t* is incubation days, *v* is the % each disease incidence on each month, and *n* is the number of estimations.

Just before harvest, samples of cherries were taken in all the test plots to make a final evaluation of damage caused by *R. cerasi* and *D. suzukii*, and thus to determine the effectiveness of the management strategies. A minimum of 200 cherries from at least 10 different trees were taken in each plot. Fruit were examined in the laboratory under a stereomicroscope to identify the presence of damage caused by any of the two aforementioned pests.

### 2.4. Effect of IPM on Microorganism and Arthropod Biodiversity

2.4.1. Estimation of Microorganism Biodiversity

Effects of the integrated disease protocols (New IPM and Standard IPM) on biodiversity of microorganisms were estimated on flower and fruit surfaces from four pairs of plots and two areas (Extremadura and Aragon). Microorganisms were measured by quantification of total genomic DNA concentration and colony-forming units on culture media.

Quantification of Total Genomic DNA Concentration

Ten fruit or flower samples from each tree in each test plot were processed as described in Garcia-Benitez et al. [25]. The extracted DNA was amplified by PCR using the 18S rDNA and 16S rDNA genes. The primers used were ITS5 and ITS4 for filamentous fungi [24], NL1 and NL4 for yeasts [26–28], and 341F and 907R for bacteria [29,30]. Total genomic DNA concentrations of bacteria, for filamentous fungi and yeasts were estimated by band area from 1.5% agarose gels. Gels were visualized and photographed using a gel imaging system (Bio-Rad Lab., Hercules, CA, USA) and analyzed with the software ImageJ J.1.53K (http://imagej.nih.gov/ij accessed on 12 January 2022). Area of each band was measured in order to determinate DNA concentration of each sample tested. Bands of each IPM were compared in all test plots.

Quantification of Colony-Forming Units (CFUs)

Nontarget yeasts and filamentous fungi were estimated as CFUs per flower and fruit on culture media. Samples with 10 flowers or fruit from each tree in each plot were suspended in 100 mL of sterile distilled water (SDW) and shaken for 30 min at 150 rpm; 10–100-fold dilutions were made. One hundred microliter aliquots from undiluted and diluted suspensions were distributed in Petri dishes with PDA plus 0.5 g $L^{-1}$ streptomycin (PDAs). Three replicates were performed for each sample and dilution. Dishes were incubated at room temperature and after 5–7 days yeast and all other fungal colonies were counted.

2.4.2. Arthropod Abundance and Diversity Evaluation

A beating/shaking sampling method based on Marcotegui et al. [31] was performed to assess arthropod populations in the canopy of cherry trees, while ground-dwelling arthropod populations were monitored by pitfall traps, as described in García-Ruiz et al. [32]. In this last case, four pitfall traps were placed in each test plot, located in the spaces between consecutive trees in a central row, and traps were operative for 8 days. Both types of sampling were carried out monthly from March to June. However, in Extremadura, it was only possible to use the data obtained in three of the four samplings, since the abundant rains flooded the pitfall traps of the May sampling.

### 2.5. Analysis of Chemical Pesticide Residues

2.5.1. Chemicals and Reagents

Standards of acetamiprid, deltamethrin, difenoconazole, dodine, fenpyrazamine, spinosad, fluopiram, tebuconazole, sulfoxaflor, λ-cyhalothrin, spirotetramat, boscalid, pyraclostrobin and fenbuconazole were purchased from Sigma (Steinheim, Germany). Acetonitrile (HPLC Far UV grade) was obtained from Labscan (Stillogan, Co., Dublin,

Ireland). The water used for the LC mobile phase and aqueous solutions was purified with a Millipore System (Milli-Q-50 18 mΩ).

Individual stock solutions containing 500 µg mL$^{-1}$ of each analyte were prepared in pesticide-grade acetonitrile. The standard solutions were stored at 4 °C in the dark and were used to prepare more diluted standard solutions for calibration standards in matrix, in solvent and for recovery study.

### 2.5.2. Sample Preparation

According to European Commission guidance [33] 1 kg of mature cherry as field samples were collected on harvest from both, the New IPM and the Standard IPM for each plot at Extremadura and Aragon locations. After being properly labeled, samples were frozen, transported to the laboratory and stored at −20 °C until they were analyzed. While frozen, cherry stones were removed by a pitting machine and homogenized for 2 min in the food processor (Foss Iberia, Barcelona, Spain). Then, two samples of 10 ± 0.3 g of the homogenized samples were weighed into a 50 mL polytetrafluoroethylene (PTFE) centrifuge tube and 10 mL acetonitrile (1% acetic acid) was added. The tube was vigorously shaken with a multivortex mixer supplied by Benchmark Scientist (Sayreville, NJ, USA). Afterwards, QuEChERS method was employed to extract the compounds of the commodity. The mixture was shaken for 1 min and centrifuged (Eppendorf Ibérica, Hamburg, Germany). Four ml of the supernatant was transferred to a 15 mL tube where dispersive SPE was used for the cleanup of the extracts. The purified supernatant was subjected to filtration through a 0.20 µm nylon filter. Samples were analyzed in duplicate.

### 2.5.3. Apparatus and Conditions

To develop the method, matrix blank samples were spiked at two concentration levels to evaluate the recovery and precision. Calibration standards in solvent-only solutions and in blank matrix extracts were prepared at seven points of different concentrations to assess linearity from the determination coefficient ($R^2$) of a matrix-matched calibration curve. The performance of the method was estimated according to the SANCO guideline 825/00 [32].

The HPLC-MS/MS system (Agilent Technologies, Santa Clara, CA, USA) consisted of a 1200 Series liquid chromatograph, equipped with a triple-quadrupole MS/MS (Agilent 6420). Chromatographic separation was achieved with a Kinetic C18 column (2.6 µm, 100 mm × 4.6 mm) (Phenomenex, Milford, MA, USA). The mobile phase was a mixture of water with 5 mM of ammonium formate (A) and acetonitrile (B) and a gradient method for the elution of the compounds were used. The flow rate was 0.7 mL min$^{-1}$ and the injection volume was 5 µL.

Detection was achieved using a triple-quadrupole system operating in the multiple-reaction monitoring (DMRM-dynamic) mode. The positive ionization mode was used for all compounds. Data processing was performed with Agilent Mass Hunter Data Acquisition software (version V1; B.07.00, Agilent, CA, USA).

### 2.6. Data Analysis

Disease incidence, band area from 1.5% agarose gels, and CFU data were analyzed by one-way analysis of variance (ANOVA) using the statistical program Statgraphics Centurion XVI (version 16.1.03 The Plains, VI, USA). Prior to analysis, CFU data and disease incidence were log (x + 1) or arcsine (x/100) transformed, respectively, in order to improve the homogeneity of variances. When the results of the F-test were significant ($p < 0.05$), the means were compared using LSD test or Student–Newman–Keul test.

The incidence of *R. cerasi* and *D. suzukii*, expressed as percentage of attacked fruit, were compared between Standard IPM and New IPM plots with a paired samples *t*-test ($p < 0.05$) using the statistical package Statgraphics Centurion XVI.

The effect of the Standard IPM considered on the abundance and diversity of arthropods in the canopy of the cherry trees and captured in pitfall traps in cherry plots was tested by linear mixed-effects models [33,34]. The type of IPM was considered a fixed factor,

with block (each orchard in Aragon and each pair of related orchards in Extremadura) as a random factor and sampling date as a repeated measures factor. The best covariance structure for the repeated measures (date) factor was selected according to the lowest value of the Akaike and Schwarz Bayesian information criteria fit statistics [35,36]. The models were fitted using a restricted maximum likelihood estimation method. If convergence was not achieved or the final Hessian matrix was not positive definite, the random factor was removed from the model as it was identified as a redundant variable. Data were previously transformed by $\ln(x + 1)$ for normality. The significance level was always $p < 0.05$. Statistical tests were performed using the SPSS 17.0 statistical program.

In addition, principal response curve (PRC) analyses were also performed to investigate changes in taxa composition and abundance. The significance of the deviations from the line representing the Standard IPM ($y = 0$) due to the New IPM was tested using an F-type permutation test (Monte Carlo simulation) with 499 permutations. When PRCs were significant, "taxa weights" were also considered to determine treatment effects on different taxa. The responses of taxa with positive weights follow the PRC patterns, whereas those with negative weights show the opposite pattern. Taxa with weights between −0.5 and 0.5 show either a weak response or a response that is unrelated to that shown in the PRC. Data on the number of captures of each taxon were transformed to $\ln(x + 1)$ before analysis. Analyses were made using the program CANOCO 4.51 [37,38].

## 3. Results

### 3.1. Forecasting Disease Development and the Use of Decision Support Systems (DSS)

CSH and CLS incidence was estimated by each model [12,13] based on mean daily T and W (Figures S1 and S2). W values > 8 h and Tª range between 10 to 20 °C, or W > 6 h and Tª 16–20 °C were considered for CSH and CLS, respectively. Estimated BR incidence varied according to plots and the number of consecutive days in which the RH was greater than 80% for the critical period (Figure S3).

CSH estimated in both Spanish cherry areas during 2021 was below 7% in the cherry-growing season (Figure S1). CLS estimated was higher in Extremadura than Aragon orchards in 2021, except in El Saso (Aragon) orchard (Figure S2). Two CLS incidence peaks were estimated in all Extremadura's orchards and in El Saso orchard (Aragon): the first peak in April, and the second one in June. Estimated BR incidence was higher in Aragon than Extremadura in 2021 (Figure S3). BR estimated was higher than 15% in two orchards in each area, even reaching 35% in El Saso and Barrachina orchards (Aragon). Only a second peak greater than 15% was estimated in Barrachina orchard (Aragon) with the rains at the beginning of June.

### 3.2. Control Options Applied in Standard IPM and New IPM

The same cultural and monitoring practices were carried out in the Standard IPM and New IPM plots in the two cherry-growing areas (Table 2). As shown in Table 3, far fewer chemically active substances were used in the New IPM than Standard IPM plots in both cherry-growing areas. Mass trapping and *B. subtilis* application were only carried out in New IPM.

### 3.3. Disease and Pest Monitoring

The observed incidence of CSH, CLS, and BR was always lower than the estimated incidence by mathematical models (Figures S1–S4), except for El Saso (Aragon), where CSH incidence exceeded 15% (Figure S4). AUDPC for CLS, CSH, and BR observed in four pairs of plots in two cherry-growing areas were similar after Standard IPM and New IPM (Table 4). except CLS in Cerro S (Extremadura) and El Saso S (Aragon) after Standard IPM, CSH in La Cuesta S and El Saso S (Aragon) after Standard IPM, and BR in Collado N and Cañadilla N (Extremadura) after New IPM, where disease incidence was higher than New IPM and Standard IPM, respectively.

**Table 3.** Control practices applied against disease and pest, at each phenological stage of the cherry trees, in the field tests to compare different IPM strategies.

| Management | Leaf Fall | Bud Swelling (BC) | Petal Visible (D) | Full Flowering (F) | Petals Fallen (G) | Ovary Growing (H) | Green Fruit (I-J-0) | Beginning of Fruit Coloring (1–3) | Maturity (4–6) |
|---|---|---|---|---|---|---|---|---|---|
| | | | | Aragon | | | | | |
| **Standard IPM** | copper | copper | difenoconazole/ fenbuconazole | | tebuconazole dodine fluopyram spirotetramat/ sulfoxaflor/ deltamethrin | difenoconazole dodine tebuconazole fluopyram spirotetramat/ sulfoxaflor/ deltamethrin | captan pyraclostrobin boscalid λ-cihalothrin/ acetamiprid/ deltamethrin | fenbuconazole pyraclostrobin λ-cyhalothrin | pyraclostrobin boscalid |
| **New IPM** | copper | calcium polysulfide | *Bacillus subtilis* (strain QST713) | | *Bacillus subtilis* (strain QST713) sulfoxaflor | difenoconazole | mass trapping fenpyrazamine | mass trapping difenoconazole | mass trapping fenpyrazamine/ fenhexamid |
| | | | | Extremadura | | | | | |
| **Standard IPM** | copper | copper deltamethrin | | difenoconazole/ fenbuconazole tebuconazole + fluopyram | dodine acetamiprid | difenoconazole dodine spinosad + hydrolyzed protein sulfoxaflor/ deltamethrin | dodine acetamiprid/ spinosad + hydrolyzed protein | dodine tebuconazole λ-cyhalothrin + hydrolyzed protein acetamiprid/ spinosad + hydrolyzed protein | tebuconazole spinosad/ acetamiprid + hydrolysed protein |
| **New IPM** | copper | calcium polysulfide | | *Bacillus subtilis* (strain QST 714) | *Bacillus subtilis* (strain QST713) sulfoxaflor/ deltamethrin [a] | difenoconazole | mass trapping | mass trapping fenpyrazamine | mass trapping difenoconazole spinosad [b] |

[a] Only a localized treatment to some branches/trees with aphid colonies. [b] Only in one orchard to trees with unharvested fruit damaged by cracking.

**Table 4.** Area under the disease curve (AUDPC) for anthracnose (CLS by *Blumeriella jaapii*), shot-hole (CS2 by *Stigmina carpophila*) and brown rot (BR by *Monilinia* spp.) observed in Extremadura and Aragon after two integrated disease-management protocols: Standard IPM and New IPM [a].

| Area | Orchard | Management | CLS | CSH | BR |
|---|---|---|---|---|---|
| Aragon | La Loma S | Standard IPM | 0.00 | 3.78 | 0.28 |
| | La Loma N | New IPM | 0.07 | 4.20 | 0.00 |
| | La Cuesta S | Standard IPM | 0.14 | 42.30 * | 0.00 |
| | La Cuesta N | New IPM | 0.00 | 2.99 | 0.00 |
| | El Saso S | Standard IPM | 212.70 * | 899.50 * | 0.00 |
| | El Saso N | New IPM | 25.60 | 542.11 | 0.00 |
| | Barrachina S | Standard IPM | 0.01 | 1.97 | 0.00 |
| | Barrachina N | New IPM | 0.00 | 0.35 | 0.00 |
| Extremadura | Collado S | Standard IPM | 3.38 | 3.49 | 2.64 |
| | Albarizas N | New IPM | 3.17 | 37.56 | 72.36 * |
| | Cañadilla S | Standard IPM | 18.06 | 2.57 | 0.00 |
| | Cañadilla N | New IPM | 5.56 | 0.66 | 91.99 * |
| | Cerro S | Standard IPM | 145.65 * | 3.39 | 1.89 |
| | Cerro N | New IPM | 0.53 | 0.65 | 2.16 |
| | Porras S | Standard IPM | 0.51 | 0.98 | 0.00 |
| | Porras N | New IPM | 67.10 | 0.65 | 0.00 |
| SME | | | 2176.12 | 6052.11 | 1277.63 |

[a] Data are the mean of AUDPC recorded from pink blossom to harvest fruit for plots under each integrated management for each disease. Mean of 10 replicate measures. When the results of the F-test were significant ($p < 0.05$) in each column, the means were compared using the LSD (least significant difference) in each plot. SME: squared mean error. * Significantly different ($p < 0.05$).

No insecticide was applied against the European cherry fly in the New IPM plots, since the thresholds were not exceeded. Only preventive action was taken against this pest by applying mass trapping in the plots with the greatest history of damage. Final European cherry fly incidence (percentage of fruit damaged) was low in the test plots in Aragon ($\leq 0.55\%$) and somewhat higher in those of Extremadura ($\leq 2\%$). On the other hand, percentages of damaged fruit by this pest were not significantly different in the two protocols in either of the two test areas (t = 1.142, $p = 0.336$ for Aragon and t = $-0.229$, $p = 0.834$ for Extremadura) (Table 5).

**Table 5.** Final incidence of cherry fruit fly (*Rhagoletis cerasi*) in field trials to evaluate the different management strategies.

| Area | Management | NºFruit | *R. cerasi* (%) [a,b] (X ± e) |
|---|---|---|---|
| Aragon | Standard IPM | 1655 | 0.19 ± 0.13 [a] |
| | New IPM | 1630 | 0.12 ± 0.07 [a] |
| Extremadura | Standard IPM | 1526 | 0.59 ± 0.41 [a] |
| | New IPM | 1435 | 0.74 ± 0.45 [a] |

[a] Data in the column *R. cerasi* (%) are average percentages of fruit attacked. [b] Means followed by the same letter and for each area (Aragon or Extremadura) are not significantly different (paired samples *t*-test, $p < 0.05$).

No fruit attacked by *D. suzukii* were found in the inspections of fruit collected in any of the test plots until the moment of fruit maturity and no insecticide was applied until that

moment. However, in the New IPM plot of Extremadura where mass trapping was applied (Albarizas) (that with the greatest history of damage by *D. suzukii*) as well as in its control Standard IPM plot (Collado), some adult captures were recorded in the inspected traps (13.75 and 4.17 adults per trap, respectively, accumulated until commercial maturity). Later, in Albarizas plot, *D. suzukii* captures and percentage of attacked fruit increased enormously in overripe cherries that had not been harvested because they were affected by cracking. A spinosad treatment was carried out at this moment in this plot to prevent the spread of the pest.

Calcium polysulfide treatments at bud swelling, used in New IPM, generally effectively prevented black cherry aphid proliferation and only localized insecticide treatments to some branches or trees were required. In the Standard IPM plots, an insecticide treatment (deltamethrin) was applied at bud swelling and another one was necessary at the end of spring (deltamethrin, sulfoxaflor or acetamiprid) in some plots.

### 3.4. Effect of IPM on Biodiversity

No significant differences were observed in the diversity of microorganisms (filamentous fungi, yeasts and bacteria) estimated by DNA quantification between New IPM and Standard IPM in flowers (Figure S5A,B) in Extremadura and Aragon. Only significant differences ($p < 0.05$) were estimated in Cañadilla (Extremadura), where less yeasts and more filamentous fungi were quantified in New IPM than in Standard IPM (Figure S5A). Significant ($p < 0.05$) differences were also observed in La Cuesta and El Saso (Aragon), where fewer yeasts and filamentous fungi and more bacteria were quantified in New IPM compared to Standard IPM (Figure S5B). No significant differences were found in microorganisms estimated in fruit between New IPM and Standard IPM in Extremadura (Figure S5C), though significant ($p < 0.05$) differences were observed in La Cuesta and El Saso (Aragon), where more filamentous fungi and less bacteria were estimated in New IPM compared to Standard IPM (Figure S5D).

CFUs of yeasts and filamentous fungi, as well as the quantification of DNA, showed that there were no significant differences between New IPM and Standard IPM in every plot studied for both cherry areas (Extremadura and Aragon) on both flowers and pre-harvest fruit (Figure S6A–D). Only significant ($p < 0.05$) differences were estimated in La Cuesta and El Saso (Aragon). More CFU of yeast on flowers and fruit were estimated in New IPM than Standard IPM in La Cuesta (Figure S6B); while less CFU of yeast on fruit were estimated in New IPM compared to Standard IPM in El Saso (Figure S6D).

All groups of canopy-dwelling arthropods considered (total arthropods, nontarget arthropods, and natural enemies) were much more abundant in the cherry orchards of Extremadura than Aragon (Table 6). Number of families of arthropods captured was also much higher in Extremadura cherry orchards. However, no significant differences were found in any of these parameters between the two protocols in any of the test areas.

Total captures of ground-dwelling arthropods could not be compared between the two study areas because there was one more sampling in Aragon than in Extremadura. However, captures of all groups of arthropods and number of families per sample were always higher in the cherry orchards of Extremadura. On the other hand, when comparing the two protocols, significant differences were only found for natural enemies captured in the pitfall traps placed in the orchards of Aragon ($349.0 \pm 30.4$ specimens in Standard IPM vs. $443.0 \pm 28.7$ in New IPM). Taxa and numbers of arthropods captured in Standard IPM and New IPM plots, in the two areas of study are shown in Supplementary Table S1 (in the canopy of the cherry trees) and Supplementary Table S2 (in pitfall traps).

Principal response curves (Figure 1) did not show significant differences between the two protocols, neither for the natural enemies in cherry canopy nor for those on the ground of the cherry orchards, in any of the study areas.

**Table 6.** Abundance and diversity of arthropods in the canopy of the cherry trees and captured in pitfall traps in cherry plots under Standard IPM and New IPM [a].

| Management | ARAGON | | | | EXTREMADURA | | | |
|---|---|---|---|---|---|---|---|---|
| | Total Abundance | Natural Enemies | Nontarget Arthropods | Nontarget Families | Total Abundance | Natural Enemies | Nontarget Arthropods | Nontarget Families |
| | Canopy-Dwelling Arthropods | | | | | | | |
| Standard IPM | 24.8 ± 6.1 | 8.8 ± 2.5 | 18.5 ± 4.9 | 7.8 ± 0.6 | 42.3 ± 11.8 | 13.0 ± 2.2 | 33.8 ± 8.2 | 12.3 ± 1.3 |
| New IPM | 24.3 ± 8.4 | 6.0 ± 1.7 | 20.0 ± 8.3 | 6.5 ± 0.7 | 39.0 ± 3.8 | 14.3 ± 2.7 | 29.8 ± 5.4 | 15.3 ± 2.0 |
| F | 0.060 | 1.235 | 0.003 | 1.299 | 0.058 | 0.133 | 0.340 | 0.250 |
| d.f. | 1, 5.4 | 1, 13.5 | 1, 3.7 | 1, 13.3 | 1, 3.7 | 1, 6.0 | 1, 4.5 | 1, 4.5 |
| P | 0.816 | 0.286 | 0.956 | 0.275 | 0.822 | 0.728 | 0.588 | 0.880 |
| | Arthropods In Pitfall Traps | | | | | | | |
| Standard IPM | 651.8 ± 50.6 | 349.0 ± 30.4 | 629.5 ± 45.0 | 71.8 ± 1.3 | 549.3 ± 97.0 | 302.3 ± 79.7 | 458.8 ± 98.4 | 54.3 ± 5.3 |
| New IPM | 793.5 ± 89.4 | 443.0 ± 28.7 * | 724.8 ± 113.9 | 69.0 ± 1.7 | 477.0 ± 48.8 | 248.8 ± 36.0 | 390.3 ± 44.0 | 58.8 ± 2.0 |
| F | 1.705 | 33.643 | 0.223 | 1.556 | 0.443 | 0.374 | 0.168 | 0.709 |
| d.f. | 1, 5.9 | 1, 6.5 | 1, 6.0 | 1, 6.2 | 1, 6.0 | 1, 6.0 | 1, 6.0 | 1, 6.0 |
| P | 0.240 | 0.001 | 0.654 | 0.257 | 0.531 | 0.563 | 0.696 | 0.432 |

[a] Values are expressed as mean per plot ± standard error. d.f.: degrees of freedom. * Significant differences ($p < 0.05$, linear mixed-effects model).

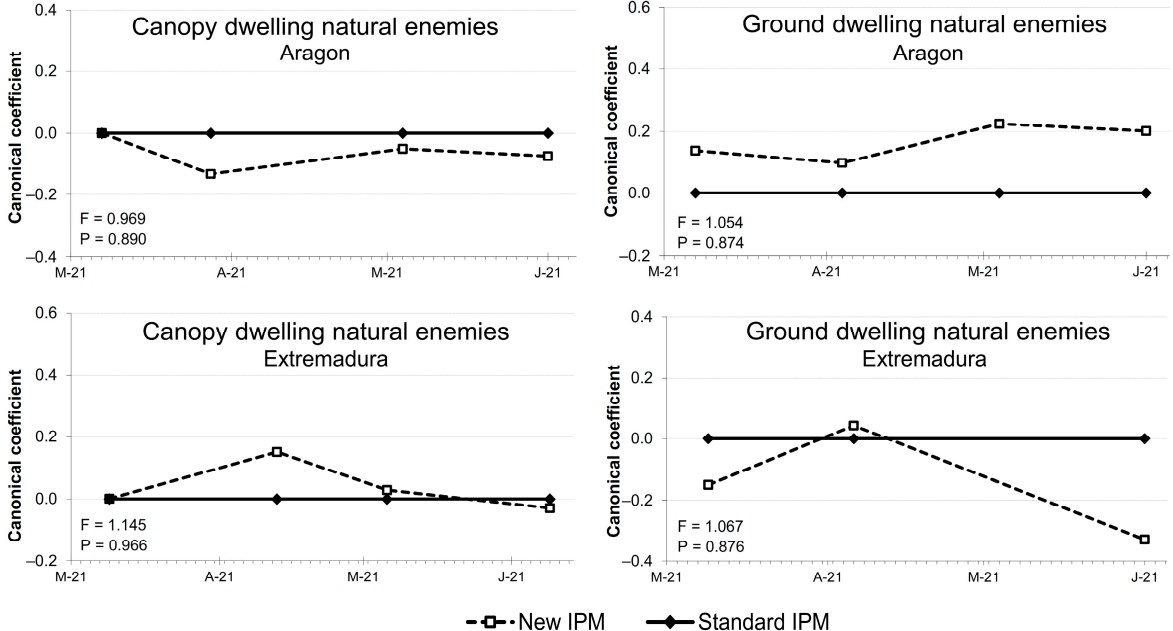

**Figure 1.** Principal response curves comparing communities of canopy and ground-dwelling natural enemies in cherry plots under Standard IPM and New IPM in two producing areas in Spain (Aragon and Extremadura). F and P: Monte Carlo simulation test results (499 permutations, $p < 0.05$). The x-axis corresponds to the sampling date.

*3.5. Analysis of Chemical Pesticide Residues*

In this work, a multiresidue method for the analysis of 14 synthetic organic pesticides has been developed for the rapid determination of residues in cherry fruit at harvest time. The insecticides and fungicides determined are the most commonly used in Spanish cherry orchards.

Considering the analytical determination of pesticide residues by HPLC-MS/MS mode, positive and negative electrospray mode were evaluated for the determination of the chemical in cherry fruit. Each of the tested analytes has two transition ions, and the most sensitive transitions in the multiple reaction monitoring (MRM) method were selected for both quantification and confirmation (Figure S7). A gradient elution program was used

since it provides a baseline resolution for the analytes, with a strong signal, good peak shapes and shortened retention time for the analytes. The performance of the method was evaluated by linearity, which was for all the tested analytes satisfactory, with determination coefficients (R2) $\geq$ 0.9978. The LOQs were considerably lower than the MRL set by the EU regulatory authorities [19], indicating that the method is sensitive enough to detect and quantify low residues levels of the tested analytes in cherry fruit. The recoveries of all the tested analytes were evaluated in blank cherry samples spiked at LOQ and they were between 74.3% and 107.4% with RSD values < 9%. The present results are consistent with the acceptable range specified by SANCO guideline 825/00 [34] of the EU on the analytical methods for pesticide residues: 70–120% and RSD < 20%.

The developed method was applied to the detection of pesticides in cherry fields following dose applications at harvest time from the two different scenarios at Extremadura and Aragon (New IPM and Standard IPM). The summary of the results obtained is shown in Table 7.

**Table 7.** Residue concentration of the pesticides applied in Aragon and Extremadura trial plots.

| Pesticide Active Substance (Class) [a] | MRL [b] (mg kg$^{-1}$) | Residue Concentration (mg kg$^{-1}$) (RSD% [c]) | | | | | | | |
|---|---|---|---|---|---|---|---|---|---|
| | | Aragon | | | | | | | |
| | | La Loma S [d] | La Loma N [d] | La Cuesta S | La Cuesta N | El Saso S | El Saso N | Barrachina S | Barrachina N |
| Acetamiprid (I) | 1.5 | - [e] | - | - | - | 0.0404 (2.85) | 0.0215 (3.79) | - | - |
| Deltametrin (I) | 0.1 | - | - | - | - | - | - | 0.0026 (3.07) | - |
| λ-cyhalotrin (I) | 0.3 | 0.0044 (1.89) | 0.0028 (4.05) | - | - | - | - | 0.0246 (1.81) | - |
| Spirotetramat (I) | 3 | 0.0016 (4.15) | - | - | - | - | - | - | - |
| Sulfoxaflor (I) | 1.5 | - | - | 0.0054 (1.95) | 0.0066 (2.32) | 0.0354 (5.24) | 0.0065 (1.04) | 0.0022 (1.15) | 0.0149 (0.90) |
| Boscalid (F) | 4 | - | - | 0.0127 (3.42) | - | - | - | - | - |
| Difenoconazole (F) | 0.3 | - | 0.0014 (3.38) | - | 0.0086 (2.48) | - | - | 0.0139 (3.06) | 0.0129 (3.98) |
| Dodine (F) | 3 | 0.0070 (1.23) | 0.0037 (2.05) | 0.0223 (2.19) | 0.0283 (3.07) | 0.0288 (2.13) | 0.0135 (0.50) | 0.0269 (4.87) | - |
| Fenbuconazole (F) | 1 | - | - | 0.0020 (2.68) | - | - | - | - | - |
| Fenpirazamide (F) | 4 | - | - | - | - | - | - | 0.0209 (1.20) | 0.0157 (2.56) |
| Fluopyram (F) | 2 | 0.2484 (2.31) | 0.2376 (3.54) | - | - | - | - | - | - |
| Pyraclostrobin (F) | 3 | - | - | 0.0109 (1.13) | - | - | - | - | - |
| | | Extremadura | | | | | | | |
| | | Collado S | Albarizas N | Cañadilla S | Cañadilla N | Cerro S | Cerro N | Porras S | Porras N |
| Acetamiprid (I) | 1.5 | 0.0104 (3.42) | - | 0.0054 (1.98) | - | 0.0086 (1.31) | - | 0.0088 (1.47) | - |
| Deltametrin (I) | 0.1 | 0.0053 (2.56) | 0.0014 (2.87) | 0.0040 (2.17) | 0.0040 (0.67) | 0.0020 (0.98) | 0.0028 (2.54) | 0.0020 (0.85) | 0.0031 (3.04) |
| λ-cyhalotrin (I) | 0.3 | 0.0290 (1.69) | - | 0.0590 (4.04) | - | - | 0.0520 (2.01) | 0.0020 (4.35) | - |

**Table 7.** *Cont.*

| Pesticide Active Substance (Class) [a] | MRL [b] (mg kg⁻¹) | Residue Concentration (mg kg⁻¹) (RSD% [c]) | | | | | | | |
|---|---|---|---|---|---|---|---|---|---|
| | | Aragon | | | | | | | |
| | | La Loma S [d] | La Loma N [d] | La Cuesta S | La Cuesta N | El Saso S | El Saso N | Barrachina S | Barrachina N |
| Sulfoxaflor (I) | 1.5 | - | 0.0396 (4.81) | - | 0.0020 (3.22) | - | - | - | - |
| Difenoconazole (F) | 0.3 | - | 0.0573 (3.37) | 0.0030 (5.21) | 0.0142 (2.95) | 0.0030 (3.25) | 0.0651 (1.65) | 0.0314 (2.16) | 0.0057 (0.68) |
| Dodine (F) | 3 | 0.0046 (3.21) | - | 0.0036 (1.8) | - | - | - | 0.0307 (3.08) | - |
| Fenpyrazamide (F) | 4 | - | 0.0429 (3.28) | - | 0.0030 (2.78) | - | 0.0172 (0.94) | - | - |
| Fluopyram (F) | 2 | 0.0025 (1.49) | - | 0.0020 (4.18) | - | 0.0030 (0.70) | - | 0.0125 (4.25) | - |
| Tebuconazole (F) | 1 | 0.0033 (0.10) | - | 0.0040 (0.78) | - | 0.0042 (3.16) | - | 0.0037 (1.02) | - |

[a] I: insecticide, F: fungicide; [b] MRL: maximum residue level [19]; [c] RSD: Relative standard deviation; [d] S: Standard IPM, N: New IPM; [e]: not detected.

In Extremadura's fields, ten pesticides (five insecticides and five fungicides) were applied considering both scenarios (Standard and New IPM) in order to control pest and diseases in each case. No residues of pesticides were found in plots where they were not applied, demonstrating that drift from adjacent plots did not occur. The number of active substances applied in each field is higher than in Aragon, e.g., in Collado and Porras Standard IMP plots, there were applied eight and nine active substances, respectively, while the maximum active substances applied in Aragon was six (Barrachina Standard IMP). A very high reduction of the number of pesticides used was observed in the New IPM Porras plot where only 2 pesticides (1 insecticide and 1 fungicide) were detected, which corresponded to an 80% reduction. This reduction does not imply a higher number of applications of both pesticides since their concentrations were 0.0031 mg kg⁻¹ and 0.0057 mg kg⁻¹ of deltametrin and difenoconazole, respectively. These concentrations were in the range of the Standard IPM, as can be observed in Table 7. In the same way, in the New IPM Collado plot the reduction in the application of pesticide were remarkably with a 50% reduction.

In Aragon's fields, the total number of pesticides applied was slightly higher than in Extremadura, 12 active substances, seven of them fungicides. However, the number of pesticides applied in each plot is in general lower than in Extremadura, as can be observed in Table 7 showing the diversity of active substances applied in each of the plots in this region. The residues detected were in the same order of magnitude of Extremadura's field and the concentrations were far below the established MRL. The highest concentration level found was fluopyram with 0.2484 mg kg⁻¹ while the MRL is 2 mg kg⁻¹. Furthermore, these results showed the rapid dissipation of the active substances when successive applications of some of the organic compounds are performed, as it did not imply a higher concentration at harvest time. In La Cuesta and Barrachina plots, there was an important reduction in the use of chemically active substances comparing Standard to New IPM till a maximum reduction of 50% in Barrachina. In El Saso and La Loma plots, the substances applied in both IPM were the same. Considering the quantities detected in both protocols, both IPM were in the same range of magnitude of mg kg⁻¹.

## 4. Discussion

Fungal diseases and pest incidence observed in the most important Spanish cherry areas were similar after Standard IPM or New IPM, despite much fewer applications of insecticides and fungicides being carried out in the New IPM, where much fewer active

substances were also used. Comparing the number of fungicides and its effectiveness in New IPM and Standard IPM technology, it was obvious that the application of fungicides against cherry diseases only when it is strictly necessary reduced the number of treatments in 2021, while good control was achieved, such as described by Borovinova and Sredkov [39] in Bulgaria. Five fungicides applied against cherry diseases in New IPM (copper, calcium polysulfide, *B. subtilis*, fenpirazamine, and difenoconazole) were sufficient and effective at the weather conditions registered in both Spanish areas in 2021. The New IPM strategy involves spraying of fungicides only when climatic conditions are conducible to fungal infection on cherry surfaces, but before germ tubes have penetrated the host tissue [40]. For performing this evaluation, preliminary knowledge about fungal infection and fungicide efficacy is necessary. New IPM cherry plots "during infection" sprays were used to better target fungicide applications against diseases, i.e., to apply chemical fungicides only when necessary in order to reduce chemicals usage during the cherry-growing season [40,41]. Application of the "during infection" sprays is supported by extensive studies of the underlying biological processes, i.e., spore deposition, germination, germ-tube elongation, formation of appressoria, and formation of penetrating hyphae [42], and by the development of mathematical models that predict the dynamics of the fungal population on plant surfaces affected by weather conditions [43,44].

Cherry diseases were estimated by mathematical models and the estimations always were higher than those diseases observed after applications in both IPM, except for one plot in each cherry area (El Saso and Cerro), where CSH incidence exceeded 15%. Cherry diseases were successfully controlled by preventive and postinfection treatments. This result confirms the findings of Borovinova [45] and Borovinova and Sredkov [39], and was similar to those obtained by Eisensmith and Jones [12,46] with sour cherry in Michigan (USA). The number of treatments against CLS was able to be reduced during the moderately wet spring in 2021. More than 5% of CLS and BR were estimated during April 2021, when the highest rainfall and the highest RH and W values were recorded. These results are in agreement with those of Borovinova [47], Holb [48], and Van Leeuven et al. [49,50], who stated that a more serious brown fruit infection of stone and pome fruit occurred in rainy periods due to fruit cracking and lower effectiveness of fungicides. BR incidence estimated was higher in Aragon than Extremadura in 2021, while CLS estimated was higher in Extremadura than Aragon orchards, except in El Saso. However, no CSH incidence of more than 5% was estimated during crop season. It is quite possible that the CSH model used does not fit the conditions of the two cherry-growing areas in Spain. CLS and BR models were a great help in forecasting disease development by predicted climatic factors and to define the needs of each of the protection measures (chemical and biological) in New IPM. New IPM minimized the number of fungicides based on disease prediction according to disease models, using mainly biological products with dry critical period, and trying in case of chemical fungicide applications an adequate management to avoid possible resistance.

No differences in damage by *R. cerasi* were observed between the IPM strategies, despite no insecticide treatment being needed in the New IPM plots, while at least one insecticide treatment was applied in those under Standard IPM. Due to the low tolerance level of damage by this pest and the local and temporal irregularity of its populations, producers tend to carry out preventive applications of insecticides, instead of using true treatment thresholds [15]. However, this practice entails environmental and residue risks and does not comply with the IPM principles [51] and thus a very exhaustive monitoring of the adults' flight is needed, even at orchard level, to carry out insecticide applications only when necessary. Mass trapping is another strategy that can help control this pest, reducing the use of insecticides [15]. In this study, mass trapping was applied to plots with the highest risk of attack, and population levels remained below the treatment threshold. However, it must be considered that, in general, *R. cerasi* populations were not high in 2021 in either areas and the action of mass trapping may be insufficient in years with high populations. In addition, this method is usually based on a high density of traps, which is

not always applicable due to its high cost [15]. In our case, effective and at the same time simple and very low-cost traps and attractants were selected, so this is not an important limitation.

Despite the general low populations of *D. suzukii* in the two study areas, in the year of the study, captures of adults were observed in plots of Extremadura near watercourses and forest masses, with high atmospheric humidity, conditions that are known to favor this pest [52]. However, in these plots, no damaged fruit was recorded before commercial maturity. This goes against the opinion of some authors according to which the most common food attractants of *D. suzukii*, such as those used in this study (wine + apple cider vinegar), are less attractive to adult females than ripening fruit, which would make them ineffective for pest monitoring and mass trapping [53]. On the other hand, in this study it was also observed that with a density of one trap/tree, mass trapping may be sufficient to defend production until commercial maturity, at least against moderate pest densities, which agrees with other experiences of mass trapping against *D. suzukii* [54].

The treatment with calcium polysulfide, applied in bud swelling in the New IPM plots, was at least as effective as the treatments with deltamethrin in the plots under Standard IPM. Calcium polysulfide or lime–sulfur solution is an inorganic chemical used intensively in the past as winter treatment to control scales, mites and some diseases, especially in fruiting trees, but its use declined with the rise of chemically synthesized pesticides. However, as lime–sulfur preparations have an apparently low ecological impact, their use in IPM systems is being recovered and it is one of the most common recommendations for controlling fungal, insect and mite infestations in organic crops [55,56]. In this study, therefore, a double benefit of the application of calcium polysulfide at bud swelling has been proven, since it has activity on both diseases and pests and replaces insecticide and fungicide with worse ecotoxicological characteristics.

Pesticides have a direct effect on microorganism diversity and harm the environment and humans. Thus, they should be used in a sustainable way [57]. Microorganisms (fungi and bacteria) colonizing cherry flowers and fruit influence plant–pathogen interactions. Therefore, the diversity and number of microorganisms on the surface of flowers and fruit are essential to prevent diseases development. No significant differences were observed between microorganism estimates in New IPM and Standard IPM on flowers or on fruit by CFUs of yeasts and filamentous fungi or the quantification of bacterial and fungal DNA, although significant differences were estimated in some plots. The main differences occurred at the yeast level in two plots in Aragon (La Cuesta and El Saso), with no similar effects in the plots in Extremadura. Then, the diversity of these microorganisms may be more controlled by climatic changes such as temperature, humidity, radiation, wind speed and rainfall than by phytosanitary treatments [58].

Although far fewer pesticide applications were carried out in New IPM plots than in those under Standard IPM, no significant differences were detected in total arthropod abundance or diversity of arthropod families, in tree canopies or on the ground, in either of the two cherry-growing areas. A longer application of the New IPM strategy is likely to be needed for a detectable recovery of the populations of these organisms. However, significantly more abundance of ground-dwelling natural enemies was observed in the New IPM plots in Aragon, although the same did not happen in those in Extremadura. The agroecosystem in the cherry-growing area of Extremadura (Jerte Valley) is much more diverse and richer, with great influence from natural areas (forest and other non-cultivated patches). This is surely the main cause of the more abundant and diverse populations of all groups of arthropods recorded in the cherry orchards of Extremadura and it also probably helped a faster recovery of natural enemies' populations after pesticide applications in the Standard IPM cherry plots in this area. On the other hand, in the Standard IPM plots in Aragon, insecticide treatments were carried out to the entire foliage of the cherry trees against the European cherry fly, while in Extremadura, bait sprays were used, which greatly reduce the amount of pesticide applied and subsequently the impact on natural enemies [59].

A multiresidue method for the analysis of 14 synthetic organic pesticides has been developed for the rapid determination of residues in cherry at harvest. This multiresidue method can be used as a practical program to monitor the level of pesticides compared to other rather complex methods involving many steps and consequently highly time- and solvent-consuming.

Regarding residue concentrations, in all cases, the quantities detected at harvest time were far below the MRL (in the order of 1 or 2 magnitude lower) both in the New IPM and in the Standard IPM. For example, the highest value found for the insecticide $\lambda$-cyhalotrin was $0.059 \text{ mg kg}^{-1}$ in Cañadilla (Extremadura) and the MRL established is $0.3 \text{ mg kg}^{-1}$. The range of concentrations found in the plots of both localities was similar to other pesticide residue concentrations found in monitoring of cherry commodity carried out by other authors, such as ppb for acetamiprid [60,61] or dodine [62].

It is important to highlight the reduction of chemical active substances in most of the trial plots comparing the New IPM to the Standard. For example, in Porras plots (Extremadura) the reduction was 75% and 50% in Barrachina. (Aragon) (Table 7).

## 5. Conclusions

To sum up, the New IPM compared to the Standard IPM applied in two Spanish regions of cherry crops showed a significant reduction in the necessity of use of active substances without compromising the effectiveness of pest and disease control. Furthermore, New IPM enhances the use of low-risk plant protection products in line with the objectives of the farm-to-fork strategy. Less use of plant protection products results in a reduction in the residue levels of pesticides in cherries, though analysis of pesticide residues in both the Standard IPM and the New IPM showed the compliance of the established MRLs. Disease prediction using models has shown to be an efficient decision tool in the New IPM that allows to take decisions on the use of fungicides. Mass trapping is a strategy that helps control pests in cherry orchards contributing to the reduction of use of insecticides. A clear benefit of the New IPM on the biodiversity of microorganisms and on the abundance and diversity of arthropods could not be appreciated in this year of study. However, increases in some groups of natural enemy arthropods were observed in the New IPM plots, and these benefits may be more evident in the longer term.

**Supplementary Materials:** The following supporting information can be downloaded at: https://www.mdpi.com/article/10.3390/agronomy12091986/s1, Figure S1: Percentage of estimated incidence of shot hole during 2021 in Extremadura and Aragon, and climatic conditions; Figure S2: Percentage of estimated incidence of leaf-spot during 2021 in Extremadura and Aragon, and climatic conditions; Figure S3: Percentage of estimated incidence of brown rot during 2021 in Extremadura and Aragon, and climatic conditions; Figure S4: Observed incidence of shot-hole, leaf-spot, and brown rot during 2021 in Extremadura and Aragon; Figure S5: Effects of the integrated disease protocols (New IPM and Standard IPM) on biodiversity of microorganisms (yeasts, filamentous fungi, and bacteria) on flowers and fruit surfaces estimated by band area from 1.5% agarose gels using the software ImageJ J.1.53K. Figure S6: Effects of the integrated disease protocols (New IPM and Standard IPM) on biodiversity of microorganisms (yeasts and filamentous fungi) on flowers and fruit surfaces estimated as log [colony-forming units (CFUs) + 1] during 2021 in Extremadura and Aragon plots; Figure S7: Mass spectra of the studied organic pesticides (a) Chromatograms of the quantification transitions at harvest (b) blank samples; Table S1: Numbers of arthropods captured in the canopy of the cherry trees in cherry orchards under Standard IPM and New IPM; Table S2: Numbers of ground-dwelling arthropods captured in pitfall traps in cherry orchards under Standard IPM and New IPM.

**Author Contributions:** Conceptualization, A.D.C., I.L., M.G.-N., I.S.-R., P.S.-E. and J.-L.A.-P.; methodology, A.D.C., I.L., M.G.-N., I.S.-R., P.S.-E., M.M.-M., G.C.; data analysis, A.D.C., I.L., M.G.-N., I.S.-R., P.S.-E. and J.-L.A.-P.; validation, M.G.-N., I.S.-R., M.M.-M. and I.L.; formal analysis, A.D.C., I.L., M.G.-N., M.M.-M., P.S.-E.; investigation, A.D.C., I.L., M.G.-N., I.S.-R., G.C., P.S.-E. and J.-L.A.-P.; resources, A.D.C., I.L., M.G.-N., I.S.-R. and J.-L.A.-P.; data curation, A.D.C., I.L., M.G.-N., I.S.-R., P.S.-E.; writing—original draft preparation, A.D.C., I.L., M.G.-N., M.M.-M., I.S.-R., P.S.-E.; writing—review

and editing, A.D.C., I.L., M.G.-N., I.S.-R., P.S.-E. and J.-L.A.-P.; visualization, A.D.C., I.L., M.G.-N., I.S.-R., P.S.-E. and J.-L.A.-P.; supervision, A.D.C., I.L., M.G.-N., I.S.-R., P.S.-E. and J.-L.A.-P.; project administration, J.-L.A.-P.; funding acquisition, A.D.C., M.G.-N. and J.-L.A.-P. All authors have read and agreed to the published version of the manuscript.

**Funding:** This study has been financed by project GO FITOSCEREZO (20190020007384), financed by the EU and 80% cofinanced by FEADER MAPA (PNDR).

**Institutional Review Board Statement:** Not applicable.

**Informed Consent Statement:** Not applicable.

**Data Availability Statement:** Not applicable.

**Acknowledgments:** We thank all members of GO FITOSCEREZO: FEPEX—Federación Española de asociaciones de Productores EXportadores de frutas, hortalizas, flores y plantas vivas; AEPLA—Asociación Empresarial para la Protección de las Plantas; DEVREG CONSULTA S.L.U. CTAEX—Centro Tecnológico Nacional Agroalimentario; ACVJ—Agrupación de Cooperativas Valle del Jerte; AEAMDE—Asociación de Empresarios Agrícolas del Margen Derecho del Ebro, and especially Alejandra Rodríguez de la Calle, Ana Delia Rodríguez Martín, Agustín Sánchez Castro, and the farmers who provided their cherry orchards for the study. We thank E. Seris, Y. Herranz and A. Cervantes for their support and collaboration from INIA, CSIC, Spain.

**Conflicts of Interest:** The authors declare no conflict of interest.

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
