# Peer review of "Development of a Disease and Pest Management Program to Reduce the Use of Pesticides in Sweet-Cherry Orchards"

_agronomy, doi:10.3390/agronomy12091986_

Round 1

Reviewer 1 Report

This study outlines a new Integrated Pest Management strategy for managing pests and diseases of sweet cherry in Spain, with the aim of reducing chemical usage. I recommend only minor changes as detailed below.

Minor comments:

33. Ton – standard abbreviation is t not Tn.

35. Add this as a reference not a weblink.

38. Specify that these are all fungi.

45. Does CSH cause damage to the fruit/crop?

52. Reword – it’s the possibility of climate factors affecting cherry diseases.

59. Is there any quantitative data on the extent of damage (yield or economic) that may be caused by these other pests?

71. Splitting this paragraph would be beneficial as it covers multiple topics - field monitoring, food monitoring, environmentally-friendly farming.

74. Not clear how monitoring levels in the field translates to 'exposure through foods'.

81. Why 50%?

87. in cherry orchards would be a better expression. 

133. Table 2 - either here or in the text, it would be worth clarifying which taxa are being classed as natural enemies, are they specific to the pests species mentioned or more general groups (e.g. parasitoids). 

175. Yeasts are fungi so some rewording needed throughout, please justify why they are singled out and whether 'fungi' then means all other fungi?

215. Can you clarify what was the total from which the cherry samples were taken - 1 kg?

217. transported not translated.

364. I presume 'higher' fungi or bacteria means more rather than a taxonomic reference (e.g. higher fungi having well-developed hyphae...etc). It might be worth clarifying the text in relation to this throughout the manuscript. 

402. I suggest changing these titles in the figure and text to canopy dwelling and epigeal or ground dwelling because currently one describes a habitat and one a collecting technique. You need to explain the x-axis, the abbreviation PRC, that the names refer to two regions...etc. Do you really mean populations or abundances?

411. This sentence does not make sense to me but perhaps it would to someone more familiar with the technique?

500. Models were a great help, contradicts previous statements in this paragraph.

Author Response

  1. Ton – standard abbreviation is t not Tn:

According to the reviewer we have corrected the sentence

  1. Add this as a reference not a weblink.

According to the reviewer the weblink has been adedd to the reference list.

  1. Specify that these are all fungi.

According to the reviewer we have corrected the sentence

  1. Does CSH cause damage to the fruit/crop?

Yes, CSH can cause damage to the crop. This has been included in the text

  1. Reword – it’s the possibility of climate factors affecting cherry diseases.

According to the reviewer we have corrected the sentence

  1. Is there any quantitative data on the extent of damage (yield or economic) that may be caused by these other pests?

We have included estimation of losses for D. suzukii, but we have not found any information for M. cerasi.

71. Splitting this paragraph would be beneficial as it covers multiple topics - field monitoring, food monitoring, environmentally-friendly farming.

According to the reviewer we have splitted the paragraph

74. Not clear how monitoring levels in the field translates to 'exposure through foods'.

We agree with the reviewer, so the sentence has been modified

  1. Why 50%?

Thank you for the comment, 50% reduction is included as a challenge in the Farm to Fork strategy, so this point has been clarified in the text

  1. in cherry orchards would be a better expression.

According to the reviewer we have corrected the sentence

  1. Table 2 - either here or in the text, it would be worth clarifying which taxa are being classed as natural enemies, are they specific to the pests species mentioned or more general groups (e.g. parasitoids).

According to the reviewer we have corrected the sentence

  1. Yeasts are fungi so some rewording needed throughout, please justify why they are singled out and whether 'fungi' then means all other fungi?

We wanted to analyse yeasts and filamentous fungi separately because PCR allowed us to do so. Now we have changed filamentous fungi to fungi throughout the manuscript.

  1. Can you clarify what was the total from which the cherry samples were taken - 1 kg?

According to the EC guidance 7029/VI/95 rev.5, 1kg is the representative field sample, so sampling was made according the cited guidance.

  1. transported not translated.

According to the reviewer we have corrected the sentence

  1. I presume 'higher' fungi or bacteria means more rather than a taxonomic reference (e.g. higher fungi having well-developed hyphae...etc). It might be worth clarifying the text in relation to this throughout the manuscript.

I am sorry, it was a grammar mistake. We are referring to numbers and not to taxonomic reference. This has now been modified throughout the whole text.

“Higher” has been replaced by “more”

  1. I suggest changing these titles in the figure and text to canopy dwelling and epigeal or ground dwelling because currently one describes a habitat and one a collecting technique. You need to explain the x-axis, the abbreviation PRC, that the names refer to two regions...etc. Do you really mean populations or abundances?

According to the reviewer we have corrected the sentence

 We have used “communities” instead of “populations” because PCR analysis considers at the same time changes in abundance and composition of taxa.

  1. This sentence does not make sense to me but perhaps it would to someone more familiar with the technique?

The sentence in the text has been improved to better describe the analytical method

  1. Models were a great help, contradicts previous statements in this paragraph.

According to the reviewer we have corrected the sentence

Models were a great help in the case of CLS and BR but however, this is not the case for CSH.

Reviewer 2 Report

Based on the research, it is not possible to compare environmental sustainability in both management strategies. So it should not be included.

Author Response

Based on the research, it is not possible to compare environmental sustainability in both management strategies. So it should not be included.

According to the reviewer we have corrected the sentence

Reviewer 3 Report

Overall a solid study with sound methodology and relevant results. The MS is well written and supported by the literature.

Some minor points:

L14 - Replace "was been" with "has been"

L34 - Is there a way to shorten this link? Or maybe include it in the reference list.

L54 - Please provide the order and families of the pest insects.

L76 - "are must to ensure". Please revise.

L82 - Please provide more information about the "farm-to-fork" strategy.

Is there a way the reviewers can access the supplementary material? They seem to be very important to the study, but I could not find them in the link you provided.

Author Response

L14 - Replace "was been" with "has been"

According to the reviewer we have corrected the sentence

L34 - Is there a way to shorten this link? Or maybe include it in the reference list.

According to the reviewer the weblink has been adedd to the reference list.

L54 - Please provide the order and families of the pest insects.

According to the reviewer we have corrected the text

L76 - "are must to ensure". Please revise.

According to the reviewer we have corrected the sentence

L82 - Please provide more information about the "farm-to-fork" strategy.

According to the reviewer we have corrected the text

Reviewer 4 Report

The paper under the title (Development of a disease and pest management program to reduce the use of pesticides in sweet cherry orchards).

-The paper discussed the application of reducing the use of chemical methods to control some common pests and diseases that affect the cherry.

-The paper is very well written

-The introduction was well presented

-The results are clearly displayed

Just one comment:

302 B. subtilis… italic.

Author Response

302 B. subtilis… italic

According to the reviewer we have corrected the text